# Association between history of HBV vaccine response and anti-SARS-CoV-2 spike antibody response to the BioNTech/Pfizer's BNT162b2 mRNA SARS-CoV-2 vaccine among healthcare workers in Japan: A prospective observational study

Momoko Iwamoto[1,2], Akira Ukimura[1,3]*, Taku Ogawa[1,3], Fumiko Kawanishi[3], Naofumi Osaka[4], Mari Kubota[4], Tatsuhiko Mori[5], Ritsuko Sawamura[5], Masami Nishihara[6], Tomio Suzuki[1], Kazuhisa Uchiyama[6,7]

1 Department of General Medicine, Osaka Medical and Pharmaceutical University Hospital, Takatsuki City, Osaka, Japan, 2 Department of Emergency and General Internal Medicine, Rakuwakai Marutamachi Hospital, Kyoto, Nakagyo Ward, Japan, 3 Infection Control Center, Osaka Medical and Pharmaceutical University Hospital, Takatsuki City, Osaka, Japan, 4 Department of Clinical Laboratory, Osaka Medical and Pharmaceutical University Hospital, Takatsuki City, Osaka, Japan, 5 Health Administration Center Osaka Medical and Pharmaceutical University, Takatsuki City, Osaka, Japan, 6 Department of Pharmacy, Osaka Medical and Pharmaceutical University Hospital, Takatsuki City, Osaka, Japan, 7 Department of General and Gastroenterological Surgery, Osaka Medical and Pharmaceutical University Hospital, Takatsuki City, Osaka, Japan

* akira.ukimura@ompu.ac.jp

## Abstract

### Introduction

Inadequate vaccine response is a common concern among healthcare workers at the frontlines of the COVID-19 pandemic. We aimed to investigate if healthcare workers with history of weak immune response to HBV vaccination are more likely to have weak responses against the BioNTech/Pfizer's BNT162b2 mRNA SARS-CoV-2 vaccine.

### Methods

We prospectively tested 954 healthcare workers for the Anti-SARS-CoV-2 spike (S) protein antibody titers prior to the first and second BNT162b2 vaccination doses and after four weeks after the second dose using Roche's Elecsys® assay. We calculated the percentage of patients who seroconverted after the first and second doses. We estimated the relative risk of non-seroconversion after the first BNT162b2 vaccine (defined as anti-SARS-CoV-2-S titer <15 U/mL) among HBV vaccine non-responders (HBs-Ab titer <10 mIU/mL) and weak responders ($\geq$10 and <100 mIU/mL) compared to normal responders ($\geq$100 mIU/mL).

### Results

Among 954 healthcare workers recruited between March 9 and March 24, 2021 at Osaka Medical and Pharmaceutical University, weak and normal HBV vaccine responders had

**Data Availability Statement:** We cannot disclose basic data to the general public, because we have not obtained consent from healthcare workers to disclose basic data to the general public and there is also the possibility of secondary use of basic data. The name of the Ethics Committee is Ethics Committee of Osaka Medical and Pharmaceutical University. The non-author's contact who can receive data access queries is Professor J Komano and his email address is jun.komano@ompu.ac.jp. He is a member of the Ethics Committee of Osaka Medical and Pharmaceutical University. The authors state that analyses can be conducted to facilitate providing information from the study to researchers.

**Funding:** The authors received no specific funding for this work.

**Competing interests:** The authors have declared that no competing interests exist.

comparable S-protein titers after the first BNT162b2 dose (51.4 [95% confidence interval 25.2–137.0] versus 59.7 [29.8–138.0] U/mL, respectively). HBV vaccine non-responders were more likely than normal responders to not seroconvert after a single dose (age and sex-adjusted relative risk 1.85 95% confidence interval [1.10–3.13]) although nearly all participants seroconverted after the second dose. After limiting the analysis to 382 patients with baseline comorbidity data, the comorbidity-adjusted relative risk of non-seroconversion among HBV vaccine non-responders to normal responders was 1.32 (95% confidence interval [0.59–2.98]).

## Discussion

Long term follow-up studies are needed to understand if protective immunity against SARS-CoV-2 wanes faster among those with history of HBV vaccine non-response and when booster doses are warranted for these healthcare workers.

## Introduction

BioNTech/Pfizer's BNT162b2 mRNA SARS-CoV-2 (COVID-19) vaccine has shown high clinical efficacy and excellent antibody response in both clinical trials and real-world settings [1–3]. More recent studies investigating the characteristics of patients with reduced humoral response to COVID-19 vaccines have shown elderly and immunosuppressed patients (i.e. cancer patients on chemotherapies, transplant recipients on immunosuppressants, hemodialysis, and high-dose glucocorticoids) to generally yield lower antibody titer responses, promoting the medical community to recommend booster doses for high-risk populations [4–7].

Understanding the duration and strength of protective immunity against COVID-19 after vaccination and being able to identify who is at risk of reduced humoral response is of paramount importance among healthcare workers who are at the frontlines of the pandemic. One understudied potential risk factor is history of reduced immune response to other vaccines such as the Hepatitis B Virus (HBV) vaccine. Vaccine non-response is well studied in HBV and is one of the few vaccines which has recommendations for serological response testing due to primary vaccine failure. Data from HBV vaccination studies show that roughly 5% of individuals are "non-responders" to HBV vaccination, meaning their immune systems do not elicit protective levels of humoral response (defined as HBs antibody titers of ≥10 mIU/mL) after receiving a full vaccination course [8]. Weak antibody response (HBs antibody titer between 10 and 100 mIU/mL) is also common, and both have been associated with older age, obesity, smoking, male gender, and immunosuppressed states [9, 10]. The mechanism for non-response is unclear, but genetic predisposition, including certain HLA allele types, and immunosenescence are thought to play a key role [11, 12].

Studies show certain immunocompromised patient groups such as hemodialysis patients and transplant recipients to have weak antibody responses after COVID-19 vaccines [7]. However, there is a dearth of studies investigated potential risk factors for diminished antibody response among generally healthy cohorts. Of particular interest among healthcare workers is knowing if robust immune response is achieved after COVID-19 vaccination for those with history of weak immune response to HBV vaccines. We therefore aimed to investigate if healthcare workers with history of non-response or weak response to HBV vaccination tend to also be weak responders to the BNT162b2 vaccine.

## Methods

Healthcare workers from Osaka Medical and Pharmaceutical University Hospital (Osaka, Japan) scheduled to receive BioNTech/Pfizer's BNT162b2 vaccine were recruited consecutively between March 9, 2021 and March 24, 2021 according to the University's vaccination prioritization schedule. Vaccination priority was given to frontline healthcare works (mostly nurses, physicians, pharmacists, technicians) and administrative workers with greater patient exposure. Among 1,051 recruited, 1,032 provided written consent to participate in the study.

We obtained baseline serum blood samples and surveys from participants immediately prior to their first vaccination dose and repeated the blood test and survey questionnaire immediately prior to and after four weeks of their second vaccination dose. We tested the blood samples using two platforms–Elecsys® Anti-SARS-CoV-2, a qualitative assay that measures the antibody responses against nucleocapsid (N) protein, and the Anti-SARS-CoV-2-S immunoassay (Roche Diagnostics International Ltd, Rotkreuz, Switzerland), a semi-quantitative assay that measures the adaptive humoral response to the SARS-CoV-2 spike (S) protein receptor binding domain. Both assays were tested on the Cobas e801 platform at our University Hospital's central laboratory. Results from Elecsys® Anti-SARS-CoV-2 (N-protein antibody) were considered positive if the cut-off index (COI) was greater than or equal to 1.0, and negative if the COI was less than 1.0. The Elecsys® Anti-SARS-CoV-2-S assay results range from 0.4 to 250 U/mL and the test was defined as "positive" if titers were 0.8 U/mL or above, and "negative" if under 0.8 U/mL [13]. The cut-off of 15 U/mL was used to define seroconversion after the BNT162b2 vaccine ("seroconverted" if titer was $\geq$15 U/mL) according to manufacturer analysis which demonstrated an inhibition cut-off of 20% on the cPass SARS-CoV-2 Neutralisation Antibody Detection Kit (Genscript, Netherlands) with a positive percent agreement of 88.9% [95% confidence interval (CI) 85.8–91.5], negative percent agreement of 90.0% [95% CI 76.3–97.2], and positive predictive value of 99.1% [95% CI 97.7–99.6] [14]. Consequently, S-protein titers between 0.8 and 15 U/mL were categorized as "weak responses."

History of HBV vaccination and anti-HBs antibody titer results were obtained from employee health records recorded by the University's occupational health program, which tests the HBV antibody titer levels of all its employees at the time of recruitment and follows their vaccination history and follow-up HBs antibody tests if they have no documented history of full vaccination. We defined HBV vaccine "non-responders" as those who had HBs antibody levels less than 10 mIU/mL after their HBV vaccination course and "weak responders" as those with titers between 10 mIU/mL and 100 mIU/mL [8, 9]. Individuals with anti-HBs titers of 100 mIU/mL or greater were defined as "normal" responders. Participants whose anti-HBs titers reached 10 mIU/mL or greater only after their second vaccine series, were categorized as a "weak responder." All participants in the study had their three-dose HBV vaccination series completed before December 2020.

We used the two-tailed Mann-Whitney test for the group comparison of antibody titer levels and the Kruskal-Wallis test for three or more group comparisons. Confidence intervals (CI) for vaccine response and other binomial proportions were calculated using the Clopper-Pearson method. Chi-square tests were used for group comparisons. Log transformation and non-parametric tests were used for non-normal data. We used relative risk (RR) to estimate the odds of not seroconverting among HBV vaccine non-responders and weak responders compared to normal responders. For this analysis, we excluded eleven patients with positive N-protein antibody test results at baseline as to exclude those with previous SARS-CoV-2 infection and calculated the age and sex-adjusted relative risk (aRR) and 95% CIs. We obtained data on smoking status, alcohol use, and comorbid conditions (history of stroke, history of cardiovascular diseases, and presence of arrhythmia, valvular heart diseases, dyslipidemia,

diabetes, hypertension, cancer, and collagen disorders) from the baseline survey to identify potential confounders. Among the subset of participants whose comorbidity data could be linked, we calculated the comorbidity adjusted relative risk of non-seroconversion by HBV vaccine response. An alpha of 0.05 was used throughout and all statistical analyses were performed using STATA (15.1, StataCorp LLC, College Station, TX) and graphics were created using R (v4.1.0). The study was approved by the University's Ethics Committee (IRB approval number 2020163).

## Results

Among 1,032 consenting healthcare workers, we excluded 71 participants from the study who had missing HBs antibody titer results and seven participants who consented but did not provide blood samples for the study. Of the 954 participants included in the analysis, the median age was 28 [IQR 34.5–45.0] years and 56% (n = 533/954) were female (Table 1). Three percent (n = 31/954) were non-responders to the HBV vaccine and 32% (n = 302/954) were weak HBV vaccine responders. At baseline, only 1% (n = 11/954) of the participants had positive antibodies against the SARS-CoV-2 N protein. Among the eleven participants with positive anti-N protein antibody results (indicating previous infection), six were unaware of their previous infection status. Of these eleven, ten also had positive test results for anti-SARS-CoV-2-S. Thirty-eight participants did not return for their SARS-CoV2-S antibody testing three weeks after their first dose of BNT162b2.

Three weeks after a single dose of BNT162b2 COVID-19 vaccine, 99.7% [95% CI 99.0–99.9] had positive SARS-CoV-2-S protein test results (titer of 0.8 U/mL and above), 85.6% [95% CI 83.3–87.7] seroconverted (titer of 15 U/mL and above), and 10.7% [95% CI 8.9–12.8] had titer levels beyond the assay range of 250 U/mL (Table 1). The median antibody titer after the first does was 56.1 (IQR 27.9–137.0) U/mL. The distribution of SARS-CoV-2 spike protein antibody titers are demonstrated in Fig 1A at baseline and after the first and second BNT162b2 COVID-19 vaccine dose. Four weeks after the second vaccination dose, all participants (n = 916/916) had positive S-protein antibodies regardless of their history for HBV vaccine response and seroconverted, and 99.3% (95% CI 98.5–99.7, n = 910/916) had titer levels above the assay range of 250 U/mL. None of the participants developed positive Anti-SARS-CoV-2 test results (positive N-protein antibody) during follow-up and no one reported being infected with SARS-CoV-2 during the study.

Increasing age was significantly associated with lower S-protein antibody titer levels after the first dose of BNT162b2 (Fig 1B, Kruskal-Wallis H test p = 0.0001). Participants under 30 years of age had a median S-protein antibody titer of 74.1 [IQR 34.6–156.0] U/mL (n = 303) and those above 50 and older had a median titer of 37.8 [IQR 15.7–86.4] U/mL (n = 160).

After a single vaccine dose, HBV non-responders were significantly less likely to seroconverge compared to normal and weak HBV vaccine responders. Among 621 normal HBV vaccine responders, 87.1% [95% CI 84.2–89.5] seroconverted after the first dose. Among 302 weak HBV vaccine responders, 84.7% [95% CI, 80.2–88.4] seroconverted after the first dose. Only 64.5% [95% CI 45.5–79.9] seroconverted after a single dose among 31 HBV vaccine non-responders (Table 1).

We found quantitative differences in the anti-SARS-CoV-2 S-protein titers across the three HBV vaccine response groups (Fig 1C, Kruskal-Wallis H test p = 0.026). More specifically, differences in the median S-protein titers were negligible between HBV vaccine responders compared to normal responders (51.4 versus 59.7 U/mL, respectively: Mann-Whitney U test, p = 0.295), but significantly lower among non-responders compared normal responders (36.9 versus 59.7 U/mL, respectively: Mann-Whitney U test, p = 0.01).

**Table 1. Characteristics and SARS-CoV-2 N and S protein antibody status before and after receiving the BNT162b2 mRNA-1273 COVID-19 vaccine among healthcare workers in the study (n = 954).**

| Characteristics | Total (n = 954) | | Normal responder (n = 621) | | Weak responder (n = 302) | | Non-responder (n = 31) | | p-values |
|---|---|---|---|---|---|---|---|---|---|
| Age (median years, IQR) | 28.0 | (34.5–45.0) | 28.0 | (35.0–44.0) | 27.0 | (33.5–45.0) | 39.5 | (49.0–54.5) | 0.0001 |
| Female sex, (n, %) | 533/954 | (56) | 361/621 | (58) | 152/302 | (50) | 20/31 | (65) | 0.050 |
| **N-protein antibody (positive n, %)** | | | | | | | | | |
| Baseline | 11/954 | (1) | 8/621 | (1) | 3/302 | (1) | 0/31 | (0) | 1.000 |
| Three weeks after 1st dose | 11/954 | (1) | 7/621 | (1) | 4/302 | (1) | 0/31 | (0) | 0.831 |
| Four weeks after 2nd dose | 10/916 | (1) | 7/594 | (1) | 3/294 | (1) | 0/28 | (0) | 1.000 |
| **Positive S-protein antibody (0.8 U/mL or above, n, %)** | | | | | | | | | |
| Baseline | 13/954 | (1) | 9/621 | (1) | 4/302 | (1) | 0/31 | (0) | 1.000 |
| Three weeks after 1st dose | 951/954 | (99.7) | 620/621 | (99.8) | 301/302 | (99.7) | 30/31 | (96.8) | 0.085 |
| Four weeks after 2nd dose | 916/916 | (100) | 594/594 | (100) | 294/294 | (100) | 28/28 | (100) | N/A |
| **Seroconversion S-protein antibody 15 U/mL or above, n, %)** | | | | | | | | | |
| Baseline | 5/954 | (0.5) | 4/621 | (0,6) | 1/302 | (0.3) | 0/31 | (0) | 1.000 |
| Three weeks after 1st dose | 817/954 | (86) | 541/621 | (87) | 256/302 | (85) | 20/31 | (65) | 0.004* |
| Four weeks after 2nd dose | 916/916 | (100) | 594/594 | (100) | 294/294 | (100) | 28/28 | (100) | N/A |
| **S-protein antibody quantitative response (median titer U/mL, IQR)** | | | | | | | | | |
| Baseline | 0.07 | (0.07–0.08) | 0.07 | (0.07–0.08) | 0.07 | (0.07–0.08) | 0.08 | (0.07–0.08) | 0.0044* |
| Three weeks after 1st dose | 56.1 | (27.9–137.0) | 59.7 | (29.8–138.0) | 51.4 | (25.2–137.0) | 36.9 | (9.7–96.7) | 0.0263* |
| Four weeks after 2nd dose | 250 | (250–250) | 250 | (250–250) | 250 | (250–250) | 250 | (250–250) | 0.0734 |

Numbers indicate median (IQR: interquartile range), n/N (%, 95% CI: confidence intervals) for outcomes. p-values for two-sided test of significance using Kruskal-Wallis H test for continuous variables and Fisher's exact and Chi-squared tests for categorical variables. Normal HBV vaccine responders are those with HBs antibody titers ≥100 mIU/mL. Weak HBV vaccine responders are those with HBs antibody titers between 10 and 100 mIU/mL. HBV vaccine non-responders are those with less than 10 mIU/mL HBs antibody response. Positive Anti-SARS-CoV-2 N-protein antibody results if COI (cut-off index) was ≥1.0.

*Denotes statistical significance at alpha = 0.05. N = nucleocapsid. S = spike.

After a single vaccine dose, HBV vaccine non-responders were at greater risk of not seroconverting (unadjusted RR 2.75, [95% CI 1.64–4.62], p<0.001) after adjusting for age and sex (aRR 1.88, [95% CI 1.13–3.18], p = 0.018, Table 2).

Among a subset of 943 participants with negative SARS-CoV-2 N antibody at baseline, we identified 382 participants whose relative risk calculation could be adjusted for by their comorbidity status obtained from the baseline survey. Among 943 participants, 188 failed to respond or responded anonymously to the survey and 373 additional were excluded who had partially or completely missing responses regarding their comorbidity status. Using the data from the remaining 382 patients and adjusting for age, sex, and past or current history of any of the

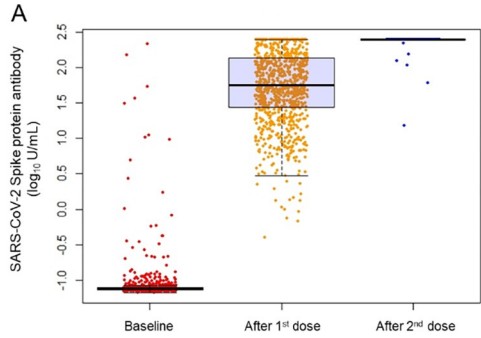

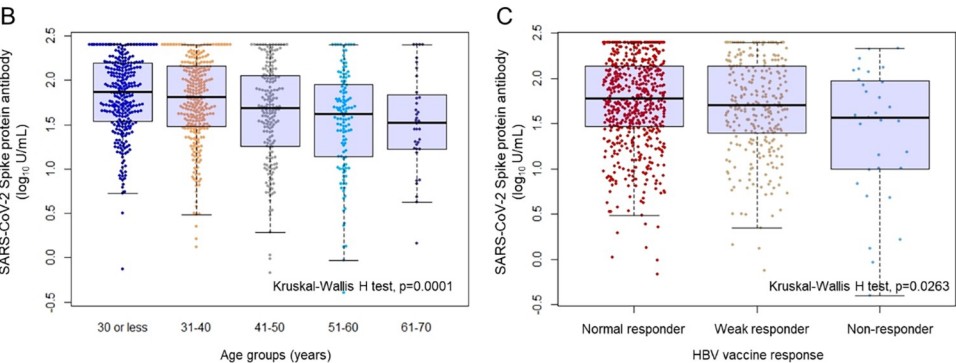

**Fig 1. Spike protein antibody levels of patients in the study.** (A) Distribution of SARS-CoV-2 spike protein antibody titers (logarithmic scale) at baseline, after one dose of BNT162b2 mRNA-1273 COVID-19 vaccine, and after two doses (n = 954). (B) Distribution of SARS-CoV-2 spike protein antibody titers (logarithmic scale) after a single dose of BNT162b2 mRNA-1273 COVID-19 vaccine, stratified by age groups (n = 954). (C) Distribution of SARS-CoV-2 spike protein antibody titers (logarithmic scale) after a single dose of BNT162b2 mRNA-1273 COVID-19 vaccine, stratified by history of HBV vaccine response (n = 954).

following: stroke, cardiovascular diseases, arrythmia, hypertension, dyslipidemia, and diabetes, we did not find HBV vaccine response to be associated with non-seroconversion risk after a single dose of BNT162b2 mRNA-1273 COVID-19 vaccine (Table 3). We did not find smoking status, alcohol status, history of cancer, collagen disorders, chronic kidney disease, respiratory disorders, and psychiatric diseases to be a potential confounder (no association to the exposure or outcome).

**Table 2. Log-binomial regression for the unadjusted and adjusted relative risk non-seroconversion (anti-SARS-CoV-2 spike protein antibody level of <15 U/mL) after a single dose of BNT162b2 mRNA-1273 COVID-19 vaccine by history of HBV vaccine response among participants with negative SARS-CoV-2 N antibody at baseline (n = 943).**

| Response to HBV vaccine | S-protein titer of <15U/mL | | Unadjusted relative risk | | Adjusted relative risk | |
|---|---|---|---|---|---|---|
| | n/N (%, [95% CI]) | | Estimate, 95% CI | p-value | Estimate, 95% CI | p-value |
| Normal responder | 80/621 | 12.9% (10.5–15.8) | 1 | – | 1 | – |
| Weak responder | 46/302 | 15.2% (11.6–19.8) | 1.18 (0.85–1.65) | 0.328 | 1.17 (0.84–1.63) | 0.357 |
| Non-responder | 11/31 | 35.5% (20.1–54.5) | 2.75 (1.64–4.62) | <0.001* | 1.88 (1.11–3.18) | 0.018* |

Weak HBV vaccine responders were those with HBs antibody titers between 10 and 100 mIU/mL. Non-responders were those with less than 10 mIU/mL HBs antibody response. Adjusted relative risk adjusted for age and sex.

*Denotes statistical significance.

**Table 3. Log-binomial regression for the unadjusted and adjusted relative risk non-seroconversion (anti-SARS-CoV-2 spike protein antibody level of <15 U/mL) after a single dose of BNT162b2 mRNA-1273 COVID-19 vaccine by history of HBV vaccine response among a subset of patients with baseline comorbidity data (n = 382).**

| Response to HBV vaccine | S-protein titer of <15U/mL | | Unadjusted relative risk | | Adjusted relative risk | |
|---|---|---|---|---|---|---|
| | n/N (%, [95% CI]) | | Estimate, 95% CI | p-value | Estimate, 95% CI | p-value |
| **Normal responder** | 36/246 | 14.6% (10.7–19.7) | 1 | – | 1 | – |
| **Weak responder** | 21/118 | 17.8% (11.8–25.9) | 1.23 (0.86–1.77) | 0.257 | 1.13 (0.70–1.83) | 0.608 |
| **Non-responder** | 5/18 | 27.8% (11.2–53.9) | 2.66 (1.54–4.60) | <0.001* | 1.32 (0.59–2.98) | 0.496 |

Weak HBV vaccine responders were those with HBs antibody titers between 10 and 100 mIU/mL. Non-responders were those with less than 10 mIU/mL HBs antibody response. Adjusted relative risk adjusted for age, sex, and comorbidity status (past or current history of any of the following: stroke, cardiovascular diseases, arrythmia, hypertension, dyslipidemia, and diabetes).

*Denotes statistical significance.

## Discussion

In our healthy and young cohort of 954 healthcare workers, 86% seroconverted and 99.7% had positive anti-SARS-CoV-2-S test results after a single dose of BNT162b2 vaccine. Seroconversion after a single vaccination dose was less common among those with history of non-response to the HBV vaccine (65%) compared to those with history of normal and weak HBV vaccine response (87% and 85%, respectively). The age and sex adjusted relative risk of having <15U/mL SARS-CoV-2-S antibody result after a single dose of BNT162b2 compared to normal HBV vaccine responders was 1.85 [95% CI 1.10–3.13]. The difference became negligible when limiting the analysis to 382 patients with available comorbidity information (past or current history of any of the following: stroke, cardiovascular diseases, arrythmia, hypertension, dyslipidemia, and diabetes). Furthermore, after the second dose, all participants (n = 916) seroconverted, and nearly all (99%) acquired SARS-CoV-2-S antibody titers above 250 U/mL.

Our findings yielded similar findings to previous real-world reports of S-protein antibody response after BNT162b2 vaccines. Shrotri et al., reported 96.3% positive anti-SARS-CoV-2-S response of ≥0.8 U/mL three to four weeks after one dose of BNT162b2 among 3,099 participants and 99.1% two weeks after the second dose among 537 participants [15]. Eyre reported 98.9% seroconversion among 3610 healthcare workers two weeks after the first dose and 99.5% among 2720 prior after the second dose [16]. Seroconversion rate was higher in our study (100%, n = 916), most likely due to the healthy worker effect as our healthcare cohort were younger than the participants in these previous studies.

Despite its observational design, one of the strengths of this study is the uniqueness of the occupational health environment in Japan where many institutions offer its employees screening for HBV at the time of recruitment, and additional dose series and follow-up antibody tests are given to ensure antibody levels reach the 10 mIU/mL cut-off. This allowed us to investigate the potential association between history of HBV vaccine non-response and response to COVID-19 vaccine which has never been studied before.

One of the limitations of this study was in the retrospective collection of occupational health records which were missing in 6.9% (n = 71/1,032) of the participants who initially consented but were later dropped from the study. These tended to be older employees (median age of 47). Misclassification may also have occurred during the study due to the retrospective nature of the occupational health data collected. HBs antibody screening is conducted for all employees at the time of recruitment and a full vaccination series is offered if their titers are below the cut-off even if they have been vaccinated. Therefore, those who developed positive HBs antibody titer only after two or more series may have been categorized as a normal HBV vaccine

responder if records of previous vaccinations were not entered into the hospital's occupational health database. However, we accepted this bias as it should theoretically lead to weaken the strength of the association towards the null.

Another limitation was the use of the cut-off value of 15 U/mL for the definition of "seroconversion" in our study. The clinical significance of "seroconversion" and the choice of the 15 U/mL threshold is arbitrary as vaccine efficacy changes over time depending on the circulating COVID-19 variant.

For our comorbidity-adjusted relative risk calculation, we dropped roughly 60% of the participants due to missing or unlinkable information. Comorbidity data was collected from survey response and not based on screening tests. The definition of each comorbidity was also not clearly defined in the survey. For example, participants were asked if they have previously been diagnosed with or are currently being treated for diabetes, but no clear definition of diabetes such as their HbA1c value was provided. Therefore, the reliability of the comorbid data in our study may be limited.

Overall, having a history of weak or non-response to the HBV vaccine did not appear to impact seroconversion of anti-SARS-CoV-2-S antibody after the second BNT162b2 vaccine dose. HBV vaccine non-responders were at greater risk of not seroconverting after a single vaccine dose compared to normal vaccine responders. After adjusting for comorbid conditions, we found the strength of the association to disappear. Our findings lead us to question if HBV vaccine non-responders have some intrinsic immunological ineptness to respond fully to COVID-19 mRNA vaccines, but insight from previous study of non-responders to HBV vaccines and tick-borne encephalitis vaccines by Garner-Spitzer et al. suggest otherwise [17]. The authors suggest that in immunocompetent individuals, non-responsiveness to a certain vaccine is most likely an antigen/ vaccine specific phenomenon and not an individual intrinsic tendency [17, 18]. Recent studies have shown comorbid conditions such as poor glycaemic control, hypercholesterolaemia, and immunosuppressed state to impact immune response to COVID-19 vaccines [19], which are also factors associated with diminished immune response to HBV vaccines. At this point, our study can only propose a potential association.

In HBV vaccination, protection is generally achieved if vaccination is done at an early age. Even if antibody titers wane over time even for those who once had very high antibody titer levels, this is not indicative of loss of protection as the host immune system is able to respond in time after exposure to HBV through the activation of memory immune cells before an infection is established [20]. As such, international guidelines no longer recommend the need for booster doses once protection is achieved after the first vaccine course [21]. For COVID-19, the exact immunological mechanism of protection from COVID-19 infection after mRNA vaccination is still being investigated. We know from recent studies that SARS-CoV-2 antibody levels and vaccine efficacy decline over time both after infection and vaccination, but that some level of immunity is sustained through both memory B cell and T cells [22–25]. Although higher anti-S protein antibody titers have been suggested as important precursors for the strength and duration of immunity [26], real-world vaccine efficacy depends also on innate immunity and external factors such as the type of variants that are circulating and people's behavior patterns and their level of daily exposure to COVID-19. However, emerging evidence clearly demonstrate declining efficacy over time [27].

Questions remain regarding what clinical significance having initial lower titer response after the first BNT162b2 dose may have, especially when all participants eventually seroconvert after the second dose. Other antibody measurement studies conducted on immunosuppressed patients also follow the same pattern–significantly lower anti-SARS-CoV-2-S antibody levels after the first dose compared to healthy cohorts, but with great majority reaching protection after the second dose [28]. Immunological research suggest different mechanisms of

protection are playing a key role in neutralization after a single or two doses [21]. Seroconversion does not necessarily mean neutralizing or immunity from SARS-CoV-2. We will need to better understand how immunity is maintained after mRNA vaccination. Longer follow-up is needed to determine if anti-SARS-CoV-2-S antibody titers also wane faster among weak or non-responders to the HBV vaccine and if that would also indicate waning protection against COVID-19 infection.

Certain patient groups such as solid-organ transplant recipients have already been identified as target groups of third booster doses [29]. Understanding who will require additional doses will not only allow us to identify healthy frontline healthcare workers at greater risk of immunological decay, but also protect immunocompromised patients in healthcare settings vulnerable to breakthrough infections that may occur among vaccinated healthcare professionals.

## Conclusion

We found healthcare workers with history of non-response to HBV vaccination to be at greater risk of not seroconverting (<15U/mL anti-SARS-CoV-2-S antibody titer) after a single dose of BNT162b2 vaccine, although all achieved seroconversion levels after the second dose. Future studies are warranted to understand if the observed effect is due to comorbid conditions that predisposes people to weaker vaccine response and if immunity against COVID-19 decays faster among HBV vaccine non-responders.

## Acknowledgments

The authors would like to thank our doctors and professors at the Osaka Medical and Pharmaceutical University Hospital, especially Yusuke Kusaka, Takashi Nakano, Ken Ogura, Kenta Minami, and Tomoyuki Yamada for their support in recruiting the study participants. We thank our secretaries Asako Yoshida and Yoko Kaide for their administrative support.

## Author Contributions

**Conceptualization:** Momoko Iwamoto, Akira Ukimura.

**Data curation:** Momoko Iwamoto.

**Formal analysis:** Momoko Iwamoto.

**Investigation:** Momoko Iwamoto, Akira Ukimura, Taku Ogawa, Fumiko Kawanishi, Naofumi Osaka, Mari Kubota, Tatsuhiko Mori, Ritsuko Sawamura, Masami Nishihara, Tomio Suzuki, Kazuhisa Uchiyama.

**Methodology:** Momoko Iwamoto, Akira Ukimura.

**Project administration:** Akira Ukimura, Tomio Suzuki, Kazuhisa Uchiyama.

**Resources:** Akira Ukimura, Tomio Suzuki, Kazuhisa Uchiyama.

**Software:** Momoko Iwamoto.

**Supervision:** Akira Ukimura, Tomio Suzuki.

**Validation:** Momoko Iwamoto, Akira Ukimura, Tomio Suzuki.

**Visualization:** Momoko Iwamoto.

**Writing – original draft:** Momoko Iwamoto.

**Writing – review & editing:** Momoko Iwamoto, Akira Ukimura, Tomio Suzuki.

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
