## [Decision Letter · Decision Letter 0]

2 Dec 2021

PONE-D-21-33288Association between history of HBV vaccine response and neutralizing antibody response to the BioNTech/Pfizer’s BNT162b2 mRNA SARS-CoV-2 vaccine among healthcare workers in Japan:A prospective observational studyPLOS ONE

Dear Dr. Ukimura,

Thank you for submitting your manuscript to PLOS ONE. After careful consideration, we feel that it has merit but does not fully meet PLOS ONE’s publication criteria as it currently stands. Therefore, we invite you to submit a revised version of the manuscript that addresses the points raised during the review process.

We look forward to receiving your revised manuscript.

Kind regards,

Jason T. Blackard, PhD

Academic Editor

PLOS ONE

Journal Requirements:

Additional Editor Comments:

This is a study of COVID-19  and HBV vaccine responses in health care workers in Japan.

The study question is an important one and worthy of investigation.

The timing of HBV vaccination is unclear.  The authors should provide more information about when HBV vaccination and boosters were given relative to COVID-19 vaccinations.

Reviewers' comments:

Reviewer's Responses to Questions

**Comments to the Author**

1. Is the manuscript technically sound, and do the data support the conclusions?

Reviewer #1: Yes

Reviewer #2: Partly

Reviewer #3: Yes

2. Has the statistical analysis been performed appropriately and rigorously? 

Reviewer #1: No

Reviewer #2: Yes

Reviewer #3: Yes

3. Have the authors made all data underlying the findings in their manuscript fully available?

Reviewer #1: No

Reviewer #2: No

Reviewer #3: Yes

4. Is the manuscript presented in an intelligible fashion and written in standard English?

Reviewer #1: Yes

Reviewer #2: Yes

Reviewer #3: Yes

5. Review Comments to the Author

Reviewer #1: This manuscript summarizes a laboratory-based study assessing the immune response to the BioNTech/Pfizer’s BNT162b2 mRNA SARS-CoV-2 vaccine, comparing health care workers (HCW) who had a weak or non-response to the hepatitis B vaccine (HBV) to those who had a strong/normal response. HCWs who had a weak response to the HBV vaccine (i.e., HBs-Ab titer < 10 mlU/mL) were more likely to have a non-neutralizing S-protein antibody response to the first BNT162b2 mRNA SARS-CoV-2 vaccine compared to those who had a normal response to the HBV vaccine (i.e., HBs-Ab titer > 100 mIU/mL). This difference diminished to non-significance after the second vaccine. The researchers acknowledge the study is not novel, however the findings are useful to build knowledge on risk of non-response to the BNT162b2 mRNA SARS-CoV-2 vaccine. For example, with vaccine in short supply in some countries, decision-makers may utilize such study results to priority who obtains a second dose first. Additionally, as the authors stated, prior vaccine response may be informative on whose protective immunity against SARS-CoV-2 wanes faster.

The odds ratios overestimate the relative risks. Thus, relative risks need to be estimated directly.

Minor comments:

71 Consider changing “promoting the scientific” to “prompting the medical”

81 Consider adding the word “male” before “gender” given the factors are specific to higher risk.

86 Delete “to”

87 Instead of “no previous studies” maybe say there is a “dearth on studies” as no studies would be difficult to document

given the vast among of publications on SARS-CoV-2.

How many HCWs were invited into the study?

147 “protein antibodies and six were unaware of their previous COVID-19 infection” needs editing. This sentence is unclear.

167 The comma after “302” needs to be removed.

191 “in our study population was slightly higher in our study” needs to be edited.

214 “We were not able to adjust for these potential confounders in our analysis as 12% (n=117/968) responded anonymously to the survey…” While it is unlikely that it will change the final conclusions of the study, you could assess confounding based on a sub analysis of those HCWs whose surveys can be linked to laboratory data.

Table 1 is difficult to read. Consider landscape (that is, rotating the page) so the information in table cells do not wrap around. Also, column labels are needed.

Table 2 (Analysis): The odds ratios overestimate the relative risks. Thus, relative risks need to be estimated directly.

Unavailability of HCW data is acceptable due to privacy concerns. The authors state that analyses can be conducted to facilitate providing information from the study to researchers.

Reviewer #2: This manuscript reports on an interesting study aimed at finding a way to predict which individuals may be at-risk for a limited response to SARS-CoV-2 mRNA vaccination by using data related to HBV vaccination response history. At a stage in the pandemic where COVID-19 vaccine boosting is being considered and durability of vaccine response is uncertain, such an approach is intriguing. There is a major limitation to the analysis and interpretation of the results, however, as the threshold for determining “neutralizing” antibodies, which the authors discuss in great detail, is of unclear origin. This must be resolved, along with a response to the other comments noted below, before this reviewer would deem this manuscript acceptable for publication.

Major Comments:

1. In the referenced article by Rubio-Acero et al., the authors find that the Roche Elecsys Anti-SARS-CoV-2 S [Ro-RBD-Ig-quant]) had a different threshold than used in the manuscript under review. From the article’s conclusion, which is supported by the results (Table 4 in Rubio-Acero et al): “For example, raw values above 28.67 U/mL for Ro-RBD-Ig-quant and above 49.78 U/mL for EI-S1-IgGquant, respectively, predicted virus neutralization > 1:5 in 95% of cases. We may hypothesize that when the value of the quantitative tests is above the predictive value (e.g., 95%), there is little benefit in performing NT and that this could act as a surrogate marker for neutralizing titers, e.g., after mass vaccinations or post-infection.” This paper also uses a surrogate test for neutralization (GS-cPass), but the threshold for the Ro-RBD-Ig-quant corresponding to ≥ 20% is ≥ 6.99. Furthermore, in reviewing the other reference cited for this cutoff, the Elecsys package insert, the reviewer could not find any indication of an appropriate cutoff value correlating with neutralizing antibody levels. Instead, the package insert states: “The results of this semi quantitative test should not be interpreted as an indication or degree of immunity or protection from reinfection” and “The clinical applicability of semi quantitative results is currently unknown and cannot be interpreted as an indication or degree of immunity nor protection from reinfection, nor compared to other SARS CoV 2 antibody assays.” Thus, it is unclear why a cut-off of 15 U/mL was used to define neutralizing immunity within this study and it seems inappropriate to state “according to manufacturer information” (line 119) in reference to any threshold used to suggest neutralizing antibody response. Clarifying this threshold is critical for the interpretation of the results, particularly those reported in line 164-170, and for the overall conclusion of the manuscript. It is unclear if a larger proportion of the weak or normal HBV vaccine responders would have not achieved a “neutralizing” threshold if a higher cutoff was used.

2. It is important to note the limitations of the assay used for this analysis. Specifically, the manufacturer states that “The performance of this test has not been established in individuals that have received a COVID 19 vaccine. The clinical significance of a positive or negative antibody result following COVID 19 vaccination has not been established, and the result from this test should not be interpreted as an indication or degree of protection from infection after vaccination.” While the reviewer understands that EUA-platforms are often evaluated in-house for off-label usage, it is important for readers to be aware of this noted limitation within the text of the article.

3. The use of the word “neutralizing” to describe the spike protein receptor binding domain (line 112) is inconsistent with the platform package insert and should be deleted, as it suggest the assay itself detects neutralizing antibodies

4. In line 117, and then later throughout when referring to the antibody results, the terms “positive” and “negative” should be used to be consistent with the manufacturer interpretation language found within the package insert

Minor Comments:

1. Test platform appears to be spelled incorrectly in some locations (line 114, 116)

2. “of” should be “or” in line 131… “three or more”

3. Why weren’t the 11 patients with reactive antibodies prior to vaccination excluded from the analysis? It seems that inclusion of these data in the analysis could confound the interpretation, specifically as it relates to predicting which individuals will or will not mount a strong response to vaccination based on HBV vaccine response (not based on HBV vaccine response + history of previous infection)

4. Maintain units when describing the semi-quantitative results, as use of the word “titer” along with a number without units may be confusing to readers (example: line 153 where only 56.1 is listed as the result)

5. Phrasing of lines 158-159 inconsistent with disease vs. pathogen (ie. “no one reported having COVID-19” or “no one reported being infected with SARS-CoV-2”)

6. For lines 170-175, please clarify what summary statistics is being used for the group comparison (median?)

7. As the authors note, the occupational health environment in Japan offers screening and additional dose series to ensure antibody levels reach the 10 mIU/mL cut-off. In what way may this bias the definition of “non-responders” compared to other settings? Is it possible to identify the cohort of individuals that did not initially respond to HBV vaccination, but ultimately achieved a weak response through additional dose series? This group may still be considered a “non-responder” in other healthcare settings where antibody level testing and additional dose series are not conducted. Some expanded discussion within lines 194-209 may help relate this study to other settings where the occupational health environment is different.

Reviewer #3: This is cohort analysis from Japan, where HBV vaccination is a strong part of public health. The investigators, through review of medical records of health workers getting COVID vaccine, identified an association between non-response to HBV vaccine and suboptimal response to first dose of COVID vaccine (the association didn’t hold after additional COVID vaccine doses). This is nice finding and is supported by what we know about vaccine efficacy. The implications of the finding could be more clearly stated to give the paper more impact.

Background:

• Scientific premise is good

• Could better explain why the focus on HBV vaccine response; people receive a wide range of vaccines. I think the reason is here is that its more common in clinical practice to check anti-HBs compared to antibodies to other vaccine-preventable infections (like measles, tetanus, etc.) . I suggest better justify the focus on HBV as part of background of the paper.

Methods:

• If the anti-HBs was low was HBV vaccine given together with COVID vaccine. I’m not aware of much data on co-administration of other vaccines with COVID. How was this handled in the program (the need to give COVID vaccine and HBV vaccine in weak/non reponders?

Results:

• Before presenting that non/weak responders to HBV vaccine were less likely to respond to COVID vaccine, please describe the # and % of non/weak HBV responders…so we have that context. Also, did they have a history of non/weak response in the past or was this based on repeat HBV vaccination (maybe I’m confused on the study design)..?

• This important line was very hard to read. Could you organize the sentence so that the % and the group are together “After a single vaccine dose, HBV non-responders were significantly less likely to reach neutralizing S-protein response compared to normal and weak HBV vaccine responders (87.1% [95% CI 84.2–166 89.5], 84.7% [95% CI, 80.2–88.4], 64.5% [95% CI 45.5–79.9] among 621 normal HBV vaccine 167 responders, 302, weak responders, and 31 non-responders, respectively [p=0.004]).”

Discussion:

• I think what is missing is more details on the implications of the paper on public health. Are you suggesting that occupational health programs should perhaps prioritize HCW with history of HBV non-response for COVID vaccines? Or booster doses? Should anti-HBs testing and HBV vaccine history assessment be expanded in places giving COVID vaccines to target outreach to certain individuals…to make sure they get 2nd, additional doses?

6. PLOS authors have the option to publish the peer review history of their article (what does this mean?). If published, this will include your full peer review and any attached files.

Reviewer #1: No

Reviewer #2: No

Reviewer #3: **Yes: **Michael Vinikoor

---

## [Author Response · Author response to Decision Letter 0]

23 Feb 2022

Additional Editor Comments:

The timing of HBV vaccination is unclear. The authors should provide more information about when HBV vaccination and boosters were given relative to COVID-19 vaccinations.

Thank you for the author comment. The classification of HBV vaccine response was based on HBV vaccination and their antibody titer response data that we have in the employee health records. The last vaccine HBV dose given among the participants in the study was December of 2020. The first COVID vaccine dose administered was March 9, 2021.

Reviewer #1:

This manuscript summarizes a laboratory-based study assessing the immune response to the BioNTech/Pfizer’s BNT162b2 mRNA SARS-CoV-2 vaccine, comparing health care workers (HCW) who had a weak or non-response to the hepatitis B vaccine (HBV) to those who had a strong/normal response. HCWs who had a weak response to the HBV vaccine (i.e., HBs-Ab titer < 10 mlU/mL) were more likely to have a non-neutralizing S-protein antibody response to the first BNT162b2 mRNA SARS-CoV-2 vaccine compared to those who had a normal response to the HBV vaccine (i.e., HBs-Ab titer > 100 mIU/mL). This difference diminished to non-significance after the second vaccine. The researchers acknowledge the study is not novel, however the findings are useful to build knowledge on risk of non-response to the BNT162b2 mRNA SARS-CoV-2 vaccine. For example, with vaccine in short supply in some countries, decision-makers may utilize such study results to priority who obtains a second dose first. Additionally, as the authors stated, prior vaccine response may be informative on whose protective immunity against SARS-CoV-2 wanes faster. The odds ratios overestimate the relative risks. Thus, relative risks need to be estimated directly.

Minor comments

71 Consider changing “promoting the scientific” to “prompting the medical”

Thank you for the author comment. We have edited the text accordingly.

81 Consider adding the word “male” before “gender” given the factors are specific to higher risk.

Thank you for the author comment. We have edited the text accordingly.

86 Delete “to”

Thank you for the author comment. We have rephrased the sentence for clarity.

87 Instead of “no previous studies” maybe say there is a “dearth on studies” as no studies would be difficult to document given the vast among of publications on SARS-CoV-2.

Thank you for pointing this out. We’ve corrected the overstatement.

How many HCWs were invited into the study?

We invited 1,051 HCWs into the study as indicated in the methods section.

147 “protein antibodies and six were unaware of their previous COVID-19 infection” needs editing. This sentence is unclear.

Thank you for this feedback. We’ve edited this section for better clarity.

“Among the eleven participants with positive N protein antibodies (indicating previous infection), six were unaware of their previous infection status. Of these eleven, ten also had reactive S-protein antibodies.”

167 The comma after “302” needs to be removed.

Thank you for this comment. Edits were made accordingly.

191 “in our study population was slightly higher in our study” needs to be edited.

Thank you for this comment. Edits have been to fix the redundant term.

214 “We were not able to adjust for these potential confounders in our analysis as 12% (n=117/968) responded anonymously to the survey…” While it is unlikely that it will change the final conclusions of the study, you could assess confounding based on a sub analysis of those HCWs whose surveys can be linked to laboratory data.

We identified 382 patients whose data on comorbidities could be linked to the baseline survey and presented the comorbidity adjusted relative risk in Table 3. After adjusting for the comorbidity status, the effect (relative risk) diminished to a non-significant level. However, to obtain this estimate, we dropped 561 participants from the analysis which may have significantly biased this finding. We added this concern in the discussion section. Thank you for this reviewer comment. We believe it added a much better insight into our findings.

Table 1 is difficult to read. Consider landscape (that is, rotating the page) so the information in table cells do not wrap around. Also, column labels are needed.

The table has been edited.

Table 2 (Analysis): The odds ratios overestimate the relative risks. Thus, relative risks need to be estimated directly.

We re-calculated the effects using relative risk.

Unavailability of HCW data is acceptable due to privacy concerns. The authors state that analyses can be conducted to facilitate providing information from the study to researchers.

 

Reviewer #2: 

This manuscript reports on an interesting study aimed at finding a way to predict which individuals may be at-risk for a limited response to SARS-CoV-2 mRNA vaccination by using data related to HBV vaccination response history. At a stage in the pandemic where COVID-19 vaccine boosting is being considered and durability of vaccine response is uncertain, such an approach is intriguing. There is a major limitation to the analysis and interpretation of the results, however, as the threshold for determining “neutralizing” antibodies, which the authors discuss in great detail, is of unclear origin. This must be resolved, along with a response to the other comments noted below, before this reviewer would deem this manuscript acceptable for publication.

Major Comments:

1. In the referenced article by Rubio-Acero et al., the authors find that the Roche Elecsys Anti-SARS-CoV-2 S [Ro-RBD-Ig-quant]) had a different threshold than used in the manuscript under review. From the article’s conclusion, which is supported by the results (Table 4 in Rubio-Acero et al): “For example, raw values above 28.67 U/mL for Ro-RBD-Ig-quant and above 49.78 U/mL for EI-S1-IgGquant, respectively, predicted virus neutralization > 1:5 in 95% of cases. We may hypothesize that when the value of the quantitative tests is above the predictive value (e.g., 95%), there is little benefit in performing NT and that this could act as a surrogate marker for neutralizing titers, e.g., after mass vaccinations or post-infection.” This paper also uses a surrogate test for neutralization (GS-cPass), but the threshold for the Ro-RBD-Ig-quant corresponding to ≥ 20% is ≥ 6.99. Furthermore, in reviewing the other reference cited for this cutoff, the Elecsys package insert, the reviewer could not find any indication of an appropriate cutoff value correlating with neutralizing antibody levels. Instead, the package insert states: “The results of this semi quantitative test should not be interpreted as an indication or degree of immunity or protection from reinfection” and “The clinical applicability of semi quantitative results is currently unknown and cannot be interpreted as an indication or degree of immunity nor protection from reinfection, nor compared to other SARS CoV 2 antibody assays.” Thus, it is unclear why a cut-off of 15 U/mL was used to define neutralizing immunity within this study and it seems inappropriate to state “according to manufacturer information” (line 119) in reference to any threshold used to suggest neutralizing antibody response. Clarifying this threshold is critical for the interpretation of the results, particularly those reported in line 164-170, and for the overall conclusion of the manuscript. It is unclear if a larger proportion of the weak or normal HBV vaccine responders would have not achieved a “neutralizing” threshold if a higher cutoff was used.

We have inserted the appropriate reference that supports our choice of the 15U/mL cut-off. The above figures and tables have been taken from the supplementary material in the research article by Kennedy, et al. In this analysis of 534 patient samples, the 15U/mL cut-off yielded PPV of 99.1% on cPass at 20% inhibition. However, as pointed out by the reviewer, using the term “neutralizing” to express anti-SARS-CoV-2-S titer levels above this cut-off would be an overstatement, especially given the recent rise in new SARS-CoV-2 variants that are circulating around the globe. We therefore changed the term to “seroconversion,” and mentioned this point in the limitations.

Kennedy NA, Lin S, Goodhand JR, Chanchlani N, Hamilton B, Bewshea C, et al. Infliximab is associated with attenuated immunogenicity to BNT162b2 and ChAdOx1 nCoV-19 SARS-CoV-2 vaccines in patients with IBD. Gut. 2021 Oct;70(10):1884-1893. doi: 10.1136/gutjnl-2021-324789.

2. It is important to note the limitations of the assay used for this analysis. Specifically, the manufacturer states that “The performance of this test has not been established in individuals that have received a COVID 19 vaccine. The clinical significance of a positive or negative antibody result following COVID 19 vaccination has not been established, and the result from this test should not be interpreted as an indication or degree of protection from infection after vaccination.” While the reviewer understands that EUA-platforms are often evaluated in-house for off-label usage, it is important for readers to be aware of this noted limitation within the text of the article.

Thank you for the reviewer feedback. We have tried to edit the limitations section in the discussion to rectify this inaccuracy and overstatement.

3. The use of the word “neutralizing” to describe the spike protein receptor binding domain (line 112) is inconsistent with the platform package insert and should be deleted, as it suggest the assay itself detects neutralizing antibodies

Thank you for the reviewer feedback. We have decided not to use the term “neutralizing” in the study for better consistency with manufacturer information.

4. In line 117, and then later throughout when referring to the antibody results, the terms “positive” and “negative” should be used to be consistent with the manufacturer interpretation language found within the package insert

Phrases such as “reactive” and “non-reactive” were changed to “positive” and “negative” for consistency with manufacturer language use.

Minor Comments:

1. Test platform appears to be spelled incorrectly in some locations (line 114, 116)

Thank you for pointing this out. We’ve corrected this mistake (Elecsys was misspelled as Elecys).

2. “of” should be “or” in line 131… “three or more”

Thank you for pointing this out. We’ve corrected this mistake.

3. Why weren’t the 11 patients with reactive antibodies prior to vaccination excluded from the analysis? It seems that inclusion of these data in the analysis could confound the interpretation, specifically as it relates to predicting which individuals will or will not mount a strong response to vaccination based on HBV vaccine response (not based on HBV vaccine response + history of previous infection)

We thank the reviewer for this point. We excluded the eleven patients from our analysis (relative risk calculation).

4. Maintain units when describing the semi-quantitative results, as use of the word “titer” along with a number without units may be confusing to readers (example: line 153 where only 56.1 is listed as the result)

Thank you for this feedback. All units were corrected for consistency. We corrected the units in line 153.

5. Phrasing of lines 158-159 inconsistent with disease vs. pathogen (ie. “no one reported having COVID-19” or “no one reported being infected with SARS-CoV-2”)

Corrections have been made accordingly.

6. For lines 170-175, please clarify what summary statistics is being used for the group comparison (median?)

The word “median” has been added for clarity.

7. As the authors note, the occupational health environment in Japan offers screening and additional dose series to ensure antibody levels reach the 10 mIU/mL cut-off. In what way may this bias the definition of “non-responders” compared to other settings? Is it possible to identify the cohort of individuals that did not initially respond to HBV vaccination, but ultimately achieved a weak response through additional dose series? This group may still be considered a “non-responder” in other healthcare settings where antibody level testing and additional dose series are not conducted. Some expanded discussion within lines 194-209 may help relate this study to other settings where the occupational health environment is different.

Thank you for the reviewer comment. We tried to address this point in the discussion section.

 

Reviewer #3: This is cohort analysis from Japan, where HBV vaccination is a strong part of public health. The investigators, through review of medical records of health workers getting COVID vaccine, identified an association between non-response to HBV vaccine and suboptimal response to first dose of COVID vaccine (the association didn’t hold after additional COVID vaccine doses). This is nice finding and is supported by what we know about vaccine efficacy. The implications of the finding could be more clearly stated to give the paper more impact.

Background:

• Scientific premise is good

• Could better explain why the focus on HBV vaccine response; people receive a wide range of vaccines. I think the reason is here is that its more common in clinical practice to check anti-HBs compared to antibodies to other vaccine-preventable infections (like measles, tetanus, etc.) . I suggest better justify the focus on HBV as part of background of the paper.

Thank you for the reviewer comment. We briefly added our rationale for our investigation in the introduction.

Methods:

• If the anti-HBs was low was HBV vaccine given together with COVID vaccine. I’m not aware of much data on co-administration of other vaccines with COVID. How was this handled in the program (the need to give COVID vaccine and HBV vaccine in weak/non reponders?

We did not have any participants in the study who had co-administration of COVID and HBV vaccines.

The last vaccine HBV dose given among the participants in the study was December of 2020. The first COVID vaccine dose administered was March 9, 2021.

Results:

• Before presenting that non/weak responders to HBV vaccine were less likely to respond to COVID vaccine, please describe the # and % of non/weak HBV responders…so we have that context. Also, did they have a history of non/weak response in the past or was this based on repeat HBV vaccination (maybe I’m confused on the study design)..?

Thank you for the reviewer comment. We reported the relevant numbers in the results section. The classification of HBV vaccine response was based on HBV vaccination and their antibody titer response data that we have in the employee health records. As we mentioned in the limitations section, it is possible that some participants were vaccinated previously by their previous employer in addition to the vaccine series offered by our institution. These participants would be included as normal vaccine responders even if they initially did not respond to the first vaccine series. Their data would theoretically result in our effect estimate towards the null value. 

• This important line was very hard to read. Could you organize the sentence so that the % and the group are together “After a single vaccine dose, HBV non-responders were significantly less likely to reach neutralizing S-protein response compared to normal and weak HBV vaccine responders (87.1% [95% CI 84.2–166 89.5], 84.7% [95% CI, 80.2–88.4], 64.5% [95% CI 45.5–79.9] among 621 normal HBV vaccine 167 responders, 302, weak responders, and 31 non-responders, respectively [p=0.004]).”

Thank you for the reviewer feedback. The sentence was rephrased accordingly.

Discussion:

• I think what is missing is more details on the implications of the paper on public health. Are you suggesting that occupational health programs should perhaps prioritize HCW with history of HBV non-response for COVID vaccines? Or booster doses? Should anti-HBs testing and HBV vaccine history assessment be expanded in places giving COVID vaccines to target outreach to certain individuals…to make sure they get 2nd, additional doses?

Thank you for the reviewer feedback. We’ve edited the discussion section.

---

## [Decision Letter · Decision Letter 1]

3 May 2022

Association between history of HBV vaccine response and anti-SARS-CoV-2 spike antibody response to the BioNTech/Pfizer’s BNT162b2 mRNA SARS-CoV-2 vaccine among healthcare workers in Japan: A prospective observational study

PONE-D-21-33288R1

Dear Dr. Ukimura,

We’re pleased to inform you that your manuscript has been judged scientifically suitable for publication and will be formally accepted for publication once it meets all outstanding technical requirements.

Kind regards,

Jason T. Blackard, PhD

Academic Editor

PLOS ONE

Additional Editor Comments (optional):

None

Reviewers' comments:

Reviewer's Responses to Questions

**Comments to the Author**

1. If the authors have adequately addressed your comments raised in a previous round of review and you feel that this manuscript is now acceptable for publication, you may indicate that here to bypass the “Comments to the Author” section, enter your conflict of interest statement in the “Confidential to Editor” section, and submit your "Accept" recommendation.

Reviewer #2: (No Response)

Reviewer #3: All comments have been addressed

2. Is the manuscript technically sound, and do the data support the conclusions?

Reviewer #2: Yes

Reviewer #3: Yes

3. Has the statistical analysis been performed appropriately and rigorously? 

Reviewer #2: Yes

Reviewer #3: Yes

4. Have the authors made all data underlying the findings in their manuscript fully available?

Reviewer #2: No

Reviewer #3: Yes

5. Is the manuscript presented in an intelligible fashion and written in standard English?

Reviewer #2: Yes

Reviewer #3: Yes

6. Review Comments to the Author

Reviewer #2: The authors have made strong attempts to respond to all reviewer comments. The additional information regarding the threshold used to determine a significant response is very clarifying. Although this reviewer appreciates the attempt to improve accuracy by no longer using the phrase "neutralizing antibodies," the use of "seroconversion" may not be the most appropriate replacement. This reviewer thinks it may confuse readers to have a "positive" antibody response (<0.8 U/ml but < 15 U/ml) be different from "seroconversion" (> 15 U/ml). Perhaps using either "strong responder" (which contrasts nicely with "weak responder") would work, or use similar language to the Kennedy et al. supplemental figure that describes the 15 U/ml threshold as maximizing "neutralizing potential." Instead of saying either "seroconverted" or "had neutralizing responses", either saying "had strong responses" or "had neutralizing potential" may help achieve the balance and clarity required (as long as both "strong response" or "neutralizing potential" were well-defined using the Kennedy reference, as is currently done for "seroconversion" in the authors' text.

Reviewer #3: (No Response)

7. PLOS authors have the option to publish the peer review history of their article (what does this mean?). If published, this will include your full peer review and any attached files.

Reviewer #2: No

Reviewer #3: **Yes: **Michael Vinikoor

---

## [Editor Report · Acceptance letter]

6 May 2022

PONE-D-21-33288R1 

Association between history of HBV vaccine response and anti-SARS-CoV-2 spike antibody response to the BioNTech/Pfizer’s BNT162b2 mRNA SARS-CoV-2 vaccine among healthcare workers in Japan: A prospective observational study 

Dear Dr. Ukimura:

I'm pleased to inform you that your manuscript has been deemed suitable for publication in PLOS ONE. Congratulations! Your manuscript is now with our production department. 

Kind regards, 

on behalf of

Dr. Jason T. Blackard 

Academic Editor

PLOS ONE